# DENOISING DIFFUSION RESTORATION MODELS

**Bahjat Kawar & Michael Elad**
Department of Computer Science
Technion, Haifa, Israel
{bahjat.kawar, elad}@cs.technion.ac.il

**Stefano Ermon & Jiaming Song**
Department of Computer Science
Stanford, California, USA
{tsong, ermon}@cs.stanford.edu

## 1 INTRODUCTION

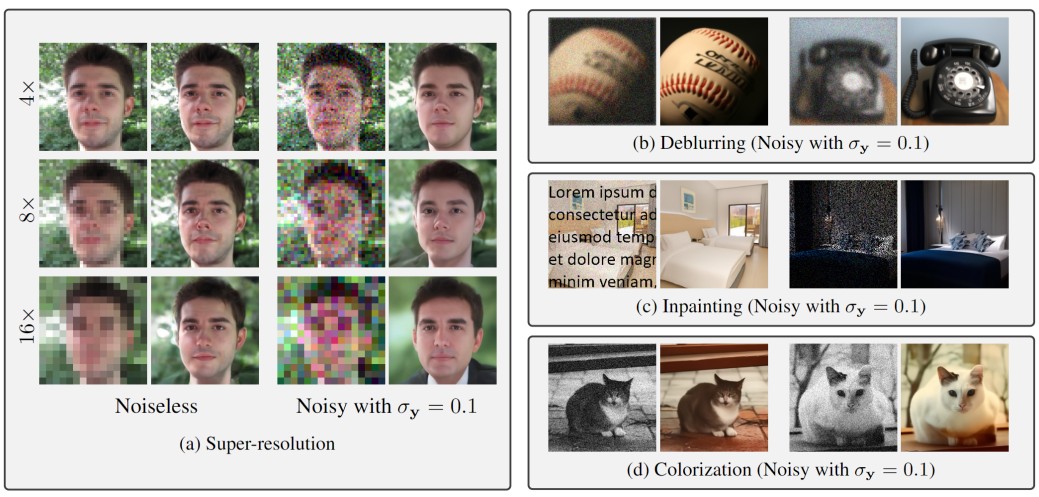

Figure 1: Pairs of measurements and recovered images with a 20-step DDRM on super-resolution, delubrring, inpainting, and colorization, with or without noise, and with unconditional generative models. The images are not accessed during training.

Many interesting tasks in image restoration can be cast as linear inverse problems. A recent family of approaches for solving these problems uses stochastic algorithms that sample from the posterior distribution of natural images given the measurements. However, efficient solutions often require problem-specific supervised training to model the posterior, whereas unsupervised methods that are not problem-specific typically rely on inefficient iterative methods. This work addresses these issues by introducing Denoising Diffusion Restoration Models (DDRM), an efficient, unsupervised posterior sampling method with generative models. Motivated by variational inference, DDRM takes advantage of a pre-trained denoising diffusion generative model for solving any linear inverse problem. We demonstrate DDRM's versatility on several image datasets for super-resolution, deblurring, inpainting, and colorization under various amounts of measurement noise. DDRM outperforms the current leading unsupervised methods on the diverse ImageNet dataset in reconstruction quality, perceptual quality, and runtime, being $5\times$ faster than the nearest competitor. DDRM also generalizes well for natural images out of the distribution of the observed ImageNet training set.

## 2 BACKGROUND

**Linear Inverse Problems.**   A general linear inverse problem is posed as

$$\mathbf{y} = \boldsymbol{H}\mathbf{x} + \mathbf{z}, \tag{1}$$

where we aim to recover the signal $\mathbf{x} \in \mathbb{R}^n$ from measurements $\mathbf{y} \in \mathbb{R}^m$. $\boldsymbol{H} \in \mathbb{R}^{m \times n}$ is a known degradation matrix, and $\mathbf{z} \sim \mathcal{N}(0, \sigma_{\mathbf{y}}^2 \boldsymbol{I})$ is an *i.i.d.* additive Gaussian noise with known variance.

The underlying structure of $\mathbf{x}$ can be represented via a generative model, denoted as $p_\theta(\mathbf{x})$. Given $\mathbf{y}$ and $\boldsymbol{H}$, a posterior over the signal can be posed as: $p_\theta(\mathbf{x}|\mathbf{y}) \propto p_\theta(\mathbf{x})p(\mathbf{y}|\mathbf{x})$, where the "likelihood"

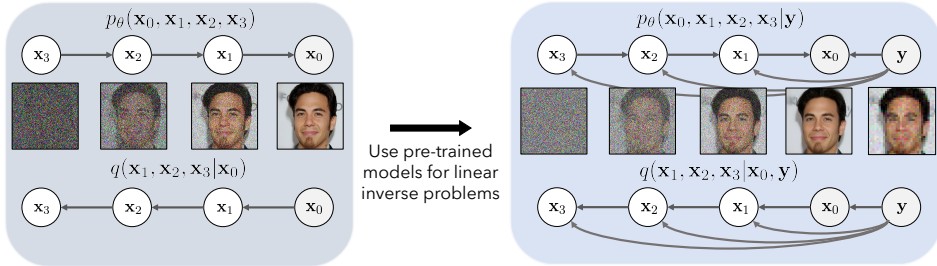

Figure 2: Illustration of our DDRM method for a specific inverse problem (super-resolution + denoising). We can use unsupervised DDPM models as a good solution to the DDRM objective.

term $p(\mathbf{y}|\mathbf{x})$ is defined via Equation 1. Recovering $\mathbf{x}$ can be done by sampling from this posterior (Bardsley, 2012), which may require many iterations to produce a good sample. Alternatively, one can also approximate this posterior by learning a model via amortized inference (*i.e.*, supervised learning); the model learns to predict $\mathbf{x}$ given $\mathbf{y}$, generated from $\mathbf{x}$ and a specific $\boldsymbol{H}$.

**Denoising Diffusion Probabilistic Models.** Structures learned by generative models have been applied to various inverse problems and often outperform data-independent structural constraints such as sparsity (Bora et al., 2017). In particular, diffusion models have demonstrated impressive unconditional generative modeling performance on images (Dhariwal & Nichol, 2021). Diffusion models (Sohl-Dickstein et al., 2015) are generative models with a Markov chain structure $\mathbf{x}_T \to \mathbf{x}_{T-1} \to \ldots \to \mathbf{x}_1 \to \mathbf{x}_0$ (where $\mathbf{x}_t \in \mathbb{R}^n$), which has the following joint distribution:

$$p_\theta(\mathbf{x}_{0:T}) = p_\theta^{(T)}(\mathbf{x}_T) \prod_{t=0}^{T-1} p_\theta^{(t)}(\mathbf{x}_t|\mathbf{x}_{t+1}).$$

After drawing $\mathbf{x}_{0:T}$, only $\mathbf{x}_0$ is kept as the sample of the generative model. To train a diffusion model, a fixed, factorized variational inference distribution is introduced:

$$q(\mathbf{x}_{1:T}|\mathbf{x}_0) = q^{(T)}(\mathbf{x}_T|\mathbf{x}_0) \prod_{t=0}^{T-1} q^{(t)}(\mathbf{x}_t|\mathbf{x}_{t+1}, \mathbf{x}_0),$$

which leads to an evidence lower bound (ELBO) on the maximum likelihood objective.

## 3 DENOISING DIFFUSION RESTORATION MODELS

Inverse problem solvers based on posterior sampling often face a dilemma: unsupervised approaches apply to general problems but are inefficient, whereas supervised ones are efficient but can only address specific problems. To this end, we introduce Denoising Diffusion Restoration Models (DDRM), an unsupervised solver for general linear inverse problems, capable of handling such tasks with or without noise in the measurements. DDRM is efficient and exhibits competitive performance compared to popular unsupervised solvers (Romano et al., 2017; Pan et al., 2020; Kawar et al., 2021). For any linear inverse problem, we define the DDRM model as

$$p_\theta(\mathbf{x}_{0:T}|\mathbf{y}) = p_\theta^{(T)}(\mathbf{x}_T|\mathbf{y}) \prod_{t=0}^{T-1} p_\theta^{(t)}(\mathbf{x}_t|\mathbf{x}_{t+1}, \mathbf{y}),$$

where $\mathbf{x}_0$ is the final diffusion output. In order to perform inference, we consider the following factorized variational distribution conditioned on $\mathbf{y}$:

$$q(\mathbf{x}_{1:T}|\mathbf{x}_0, \mathbf{y}) = q^{(T)}(\mathbf{x}_T|\mathbf{x}_0, \mathbf{y}) \prod_{t=0}^{T-1} q^{(t)}(\mathbf{x}_t|\mathbf{x}_{t+1}, \mathbf{x}_0, \mathbf{y}).$$

In the remainder of the section, we construct suitable variational problems given $\boldsymbol{H}$ and $\sigma_\mathbf{y}$ and connect them to unconditional diffusion generative models. To simplify notations, we will construct

the variational distribution $q$ such that $q(\mathbf{x}_t|\mathbf{x}_0) = \mathcal{N}(\mathbf{x}_0, \sigma_t^2\boldsymbol{I})$ for noise levels $0 = \sigma_0 < \sigma_1 < \sigma_2 < \ldots < \sigma_T$.

Now we are ready to introduce specific forms of DDRM that are suitable for inverse problems. Similar to SNIPS (Kawar et al., 2021), we consider the singular value decomposition (SVD) of $\boldsymbol{H}$, and perform the diffusion in its spectral space. The idea behind this is to tie the noise present in the measurements $\mathbf{y}$ with the diffusion noise in $\mathbf{x}_{1:T}$, ensuring that the diffusion result $\mathbf{x}_0$ is faithful to the measurements. By using the SVD, we identify the data from $\mathbf{x}$ that is missing in $\mathbf{y}$, and synthesize it using a diffusion process. In conjunction, the noisy data in $\mathbf{y}$ undergoes a denoising process. For example, in inpainting with noise (*e.g.*, $\boldsymbol{H} = \mathrm{diag}([1, \ldots, 1, 0, \ldots, 0])$, $\sigma_{\mathbf{y}} \geq 0$), the spectral space is simply the pixel space, so the model should generate the missing pixels and denoise the observed ones in $\mathbf{y}$. For a general linear $\boldsymbol{H}$, its SVD is given as

$$\boldsymbol{H} = \boldsymbol{U}\boldsymbol{\Sigma}\boldsymbol{V}^\top \tag{2}$$

where $\boldsymbol{U} \in \mathbb{R}^{m \times m}$, $\boldsymbol{V} \in \mathbb{R}^{n \times n}$ are orthogonal matrices, and $\boldsymbol{\Sigma} \in \mathbb{R}^{m \times n}$ is a rectangular diagonal matrix containing the singular values of $\boldsymbol{H}$, ordered descendingly. As this is the case in most useful degradation models, we assume $m \leq n$, but our method would work for $m > n$ as well. We denote the singular values as $s_1 \geq s_2 \geq \ldots \geq s_m$, and define $s_i = 0$ for $i \in [m+1, n]$. We use the shorthand notations for values in the spectral space: $\bar{\mathbf{x}}_t^{(i)}$ is the $i$-th index of the vector $\bar{\mathbf{x}}_t = \boldsymbol{V}^\top\mathbf{x}_t$, and $\bar{\mathbf{y}}^{(i)}$ is the $i$-th index of the vector $\bar{\mathbf{y}} = \boldsymbol{\Sigma}^\dagger\boldsymbol{U}^\top\mathbf{y}$ (where $\dagger$ denotes the Moore–Penrose pseudo-inverse). Because $\boldsymbol{V}$ is an orthogonal matrix, we can recover $\mathbf{x}_t$ from $\bar{\mathbf{x}}_t$ exactly by left multiplying $\boldsymbol{V}$. For each index $i$ in $\bar{\mathbf{x}}_t$, we define the variational distribution as:

$$q^{(T)}(\bar{\mathbf{x}}_T^{(i)}|\mathbf{x}_0, \mathbf{y}) = \begin{cases} \mathcal{N}(\bar{\mathbf{y}}^{(i)}, \sigma_T^2 - \frac{\sigma_{\mathbf{y}}^2}{s_i^2}) & \text{if } s_i > 0 \\ \mathcal{N}(\bar{\mathbf{x}}_0^{(i)}, \sigma_T^2) & \text{if } s_i = 0 \end{cases} \tag{3}$$

$$q^{(t)}(\bar{\mathbf{x}}_t^{(i)}|\mathbf{x}_{t+1}, \mathbf{x}_0, \mathbf{y}) = \begin{cases} \mathcal{N}(\bar{\mathbf{x}}_0^{(i)} + \sqrt{1-\eta^2}\sigma_t\frac{\bar{\mathbf{x}}_{t+1}^{(i)} - \bar{\mathbf{x}}_0^{(i)}}{\sigma_{t+1}}, \eta^2\sigma_t^2) & \text{if } s_i = 0 \\ \mathcal{N}(\bar{\mathbf{x}}_0^{(i)} + \sqrt{1-\eta^2}\sigma_t\frac{\bar{\mathbf{y}}^{(i)} - \bar{\mathbf{x}}_0^{(i)}}{\sigma_{\mathbf{y}}/s_i}, \eta^2\sigma_t^2) & \text{if } \sigma_t < \frac{\sigma_{\mathbf{y}}}{s_i} \\ \mathcal{N}((1-\eta_b)\bar{\mathbf{x}}_0^{(i)} + \eta_b\bar{\mathbf{y}}^{(i)}, \sigma_t^2 - \frac{\sigma_{\mathbf{y}}^2}{s_i^2}\eta_b^2) & \text{if } \sigma_t \geq \frac{\sigma_{\mathbf{y}}}{s_i} \end{cases} \tag{4}$$

where $\eta \in (0, 1]$ is a hyperparameter controlling the variance of the transitions, and $\eta$ and $\eta_b$ may depend on $\sigma_t, s_i, \sigma_{\mathbf{y}}$. We then define DDRM with trainable parameters $\theta$ as follows:

$$p_\theta^{(T)}(\bar{\mathbf{x}}_T^{(i)}|\mathbf{y}) = \begin{cases} \mathcal{N}(\bar{\mathbf{y}}^{(i)}, \sigma_T^2 - \frac{\sigma_{\mathbf{y}}^2}{s_i^2}) & \text{if } s_i > 0 \\ \mathcal{N}(0, \sigma_T^2) & \text{if } s_i = 0 \end{cases} \tag{5}$$

$$p_\theta^{(t)}(\bar{\mathbf{x}}_t^{(i)}|\mathbf{x}_{t+1}, \mathbf{y}) = \begin{cases} \mathcal{N}(\bar{\mathbf{x}}_{\theta,t}^{(i)} + \sqrt{1-\eta^2}\sigma_t\frac{\bar{\mathbf{x}}_{t+1}^{(i)} - \bar{\mathbf{x}}_{\theta,t}^{(i)}}{\sigma_{t+1}}, \eta^2\sigma_t^2) & \text{if } s_i = 0 \\ \mathcal{N}(\bar{\mathbf{x}}_{\theta,t}^{(i)} + \sqrt{1-\eta^2}\sigma_t\frac{\bar{\mathbf{y}}^{(i)} - \bar{\mathbf{x}}_{\theta,t}^{(i)}}{\sigma_{\mathbf{y}}/s_i}, \eta^2\sigma_t^2) & \text{if } \sigma_t < \frac{\sigma_{\mathbf{y}}}{s_i} \\ \mathcal{N}((1-\eta_b)\bar{\mathbf{x}}_{\theta,t}^{(i)} + \eta_b\bar{\mathbf{y}}^{(i)}, \sigma_t^2 - \frac{\sigma_{\mathbf{y}}^2}{s_i^2}\eta_b^2) & \text{if } \sigma_t \geq \frac{\sigma_{\mathbf{y}}}{s_i}. \end{cases} \tag{6}$$

where we use the symbol $\mathbf{x}_{\theta,t}$ to represent this prediction of $\mathbf{x}_0$ made by a model.

Once we have defined $p_\theta^{(t)}$ and $q^{(t)}$ by choosing $\sigma_{1:T}$, $\eta$ and $\eta_b$, we can learn model parameters $\theta$ by maximizing the resulting ELBO objective (in Appendix, Equation 8). However, this approach is not desirable since we have to learn a different model for each inverse problem (given $\boldsymbol{H}$ and $\sigma_{\mathbf{y}}$), which is not flexible enough for arbitrary inverse problems. Fortunately, this does not have to be the case. In the following statement, we show that an optimal solution to DDPM / DDIM can also be an optimal solution to a DDRM problem, under reasonable assumptions used in prior work (Ho et al., 2020; Song et al., 2021).

**Theorem 3.1.** *Assume that the models $f_\theta^{(t)}$ and $f_\theta^{(t')}$ are independent whenever $t \neq t'$, then when $\eta = 1$ and $\eta_b = \frac{2\sigma_t^2}{\sigma_t^2 + \sigma_{\mathbf{y}}^2/s_i^2}$, the ELBO objective of DDRM (details in Equation 8) can be rewritten in the form of the DDPM / DDIM objective (Song et al., 2021).*

Table 1: Noiseless $4\times$ super-resolution and deblurring results on ImageNet 1K ($256 \times 256$).

| Method | $4\times$ super-resolution | | | Deblurring | | |
|---|---|---|---|---|---|---|
| | PSNR↑ | KID↓ | NFEs↓ | PSNR↑ | KID↓ | NFEs↓ |
| Bicubic / Blurry | 25.65 | 44.90 | **0** | 19.26 | 38.00 | **0** |
| DGP | 23.06 | 21.22 | 1500 | 22.70 | 27.60 | 1500 |
| RED | 26.08 | 53.55 | 100 | 26.16 | 21.21 | 500 |
| SNIPS | 17.58 | 35.17 | 1000 | 34.32 | **0.49** | 1000 |
| DDRM | **26.55** | 7.22 | 20 | 35.64 | 7.14 | 20 |
| DDRM-CC | **26.55** | **6.56** | 20 | **35.65** | 7.03 | 20 |

Table 2: $4\times$ super resolution and deblurring results on ImageNet 1K ($256 \times 256$). Input images have an additive noise of $\sigma_{\mathbf{y}} = 0.05$.

| Method | $4\times$ super-resolution | | | Deblurring | | |
|---|---|---|---|---|---|---|
| | PSNR↑ | KID↓ | NFEs↓ | PSNR↑ | KID↓ | NFEs↓ |
| Bicubic / Blurry | 22.55 | 67.86 | **0** | 18.35 | 75.50 | **0** |
| DGP | 20.69 | 42.17 | 1500 | 21.20 | 34.02 | 1500 |
| RED | 22.90 | 43.45 | 100 | 14.69 | 121.82 | 500 |
| SNIPS | 16.30 | 67.77 | 1000 | 16.37 | 77.96 | 1000 |
| DDRM | 25.21 | 12.43 | 20 | 25.45 | 15.24 | 20 |
| DDRM-CC | **25.22** | **10.82** | 20 | **25.46** | **13.49** | 20 |

## 4 EXPERIMENTS

We demonstrate our algorithm's capabilities on CelebA-HQ (Karras et al., 2018), LSUN bedrooms, and LSUN cats (Yu et al., 2015) (all $256 \times 256$ pixels), as well as ImageNet $256 \times 256$ and $512 \times 512$. Some of the ImageNet models (taken from (Dhariwal & Nichol, 2021)) require class information. For these models, we use the ground truth labels as input, and denote our algorithm as DDRM class conditional (DDRM-CC). In all experiments, we use $\eta = 0.85$, $\eta_b = 1$, and a uniformly-spaced timestep schedule based on the 1000-step pre-trained models.

We compare DDRM (with 20 and 100 steps) with other unsupervised methods that work in reasonable time (requiring 1500 NFEs or less) and can operate on ImageNet. Namely, we compare with RED (Romano et al., 2017), DGP (Pan et al., 2020), and SNIPS (Kawar et al., 2021). The exact setup of each method is detailed in Appendix G. In addition, we show upscaling by bicubic interpolation as a baseline for super-resolution, and the blurry image itself as a baseline for deblurring.

We evaluate all methods on $4\times$ super-resolution and deblurring, on one validation set image from each of the 1000 ImageNet classes, following (Pan et al., 2020). Table 1 shows that DDRM outperforms all baseline methods, in all metrics, and on both problems with only 20 steps. The only exception to this is that SNIPS achieves better KID than DDRM in noiseless deblurring, but it requires $50\times$ more NFEs to do so. DGP and DDRM-CC use ground-truth class labels for the test images to aid in the restoration process, and thus have an unfair advantage. DDRM's appeal compared to previous methods becomes more substantial when significant noise is added to the measurements. Under this setting, DGP, RED, and SNIPS all fail to produce viable results, as evident in Table 2.

DDRM produces high quality reconstructions across all the tested datasets and problems, as can be seen in Appendix A. As it is a posterior sampling algorithm, DDRM can produce multiple outputs for the same input, as demonstrated in Figure 7. Moreover, the unconditional ImageNet diffusion models can be used to solve inverse problems on out-of-distribution images with general content. In Figure 8, we show DDRM successfully restoring $256 \times 256$ images from USC-SIPI (Weber, 1997) that do not necessarily belong to any ImageNet class.

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

## ACKNOWLEDGEMENTS

We thank Kristy Choi and Charlie Marx for insightful discussions and feedback. This research was supported by NSF (#1651565, #1522054, #1733686), ONR (N00014-19-1-2145), AFOSR (FA9550-19-1-0024), ARO (W911NF-21-1-0125), Sloan Fellowship, Amazon AWS, Stanford Institute for Human-Centered Artificial Intelligence (HAI), Google Cloud, the Israel Science Foundation (ISF) under Grant 335/18, the Israeli Council For Higher Education - Planning & Budgeting Committee, and the Stephen A. Kreynes Fellowship.

## A  ADDITIONAL EXPERIMENTS

We provide additional figures below showing DDRM's versatility across different datasets, inverse problems, and noise levels (Figures 3, 4, 5, 6 and 8). We also showcase the sample diversity provided by DDRM in Figure 7.

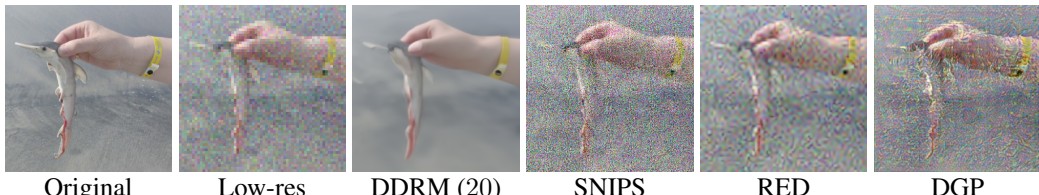

Figure 3: $4\times$ noisy super resolution comparison with $\sigma_{\mathbf{y}} = 0.05$.

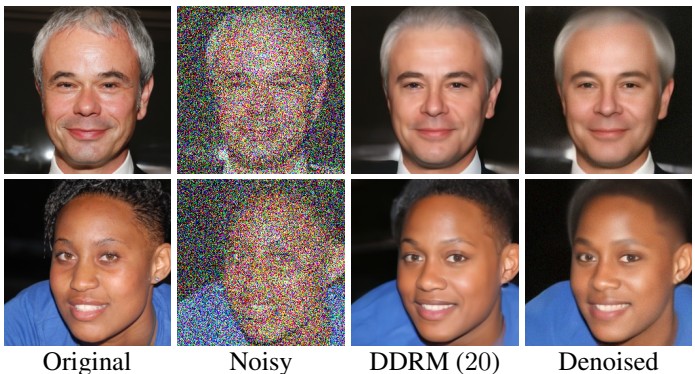

Figure 4: Denoising ($\sigma_{\mathbf{y}} = 0.75$) face images. DDRM restores more fine details (*e.g.* hair) than an MMSE denoiser.

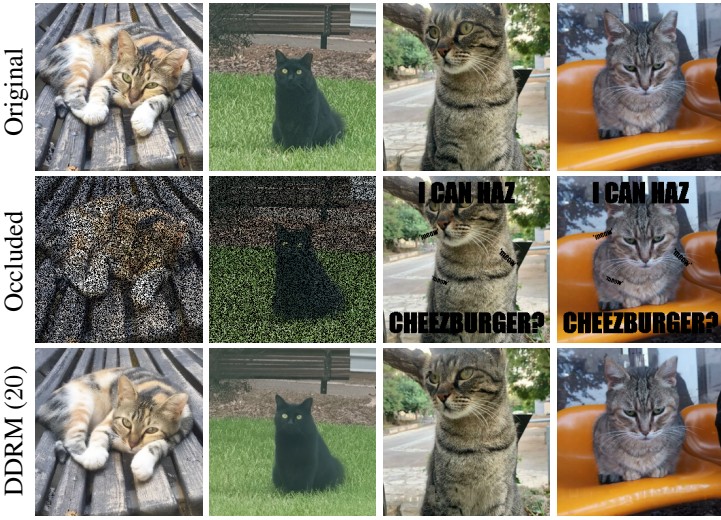

Figure 5: Inpainting results on cat images. First two images have $50\%$ of their pixels removed, last two are occluded by text.

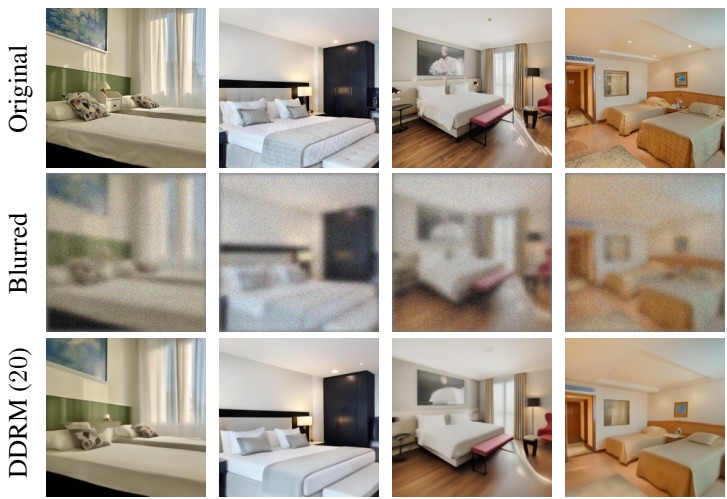

Figure 6: Deblurring results on bedroom images. Blurred images contain noise with $\sigma_{\mathbf{y}} = 0.05$.

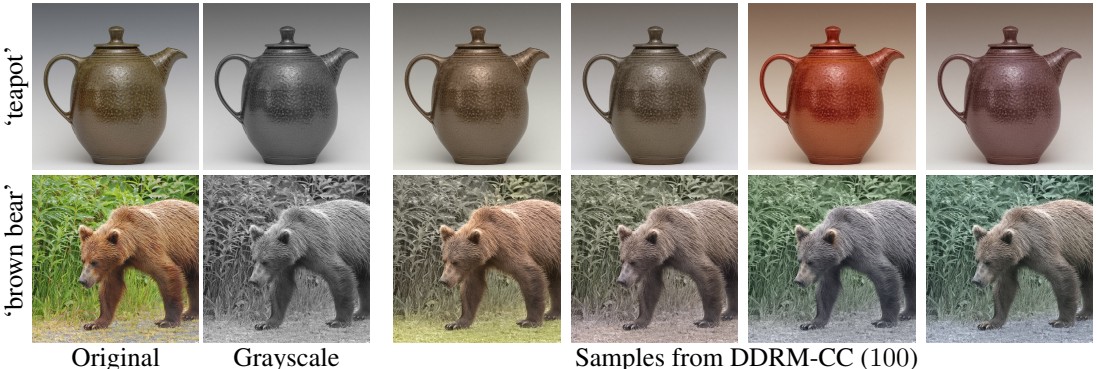

Original      Grayscale            Samples from DDRM-CC (100)

Figure 7: $512 \times 512$ ImageNet colorization. DDRM-CC produces various samples for multiple runs on the same input.

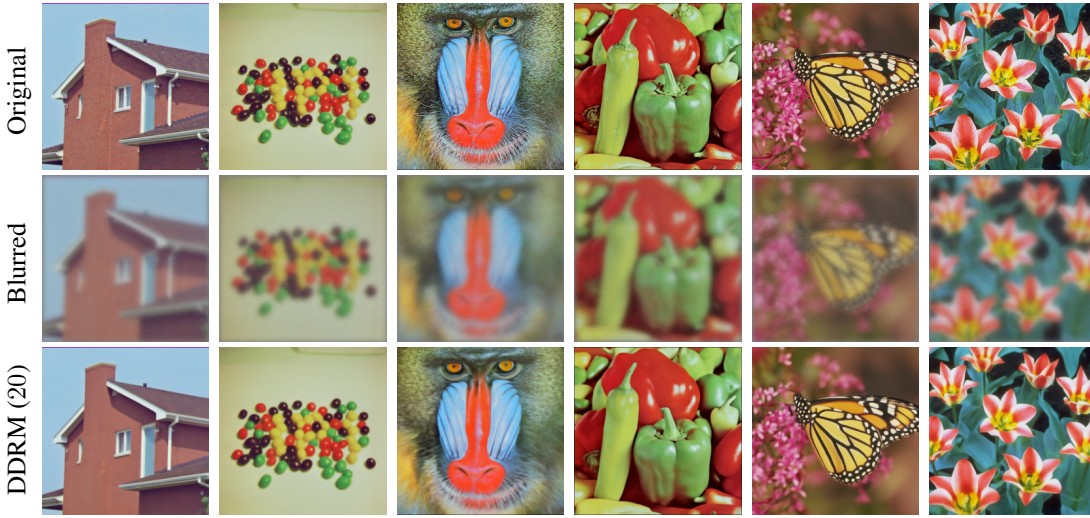

Figure 8: Results on $256 \times 256$ USC-SIPI images using an ImageNet model. Blurred images have a noise of $\sigma_{\mathbf{y}} = 0.01$.

## B DETAILS OF THE DDRM ELBO OBJECTIVE

DDRM is a Markov chain conditioned on $\mathbf{y}$, which would lead to the following ELBO objective (Song et al., 2021):

$$\mathbb{E}_{\mathbf{x}_0 \sim q(\mathbf{x}_0), \mathbf{y} \sim q(\mathbf{y}|\mathbf{x}_0)}[\log p_\theta(\mathbf{x}_0|\mathbf{y})] \tag{7}$$

$$\geq -\mathbb{E}\left[\sum_{t=1}^{T-1} D_{\mathrm{KL}}(q^{(t)}(\mathbf{x}_t|\mathbf{x}_{t+1}, \mathbf{x}_0, \mathbf{y}) \| p_\theta^{(t)}(\mathbf{x}_t|\mathbf{x}_{t+1}, \mathbf{y}))\right] + \mathbb{E}\left[\log p_\theta^{(0)}(\mathbf{x}_0|\mathbf{x}_1, \mathbf{y})\right]$$

$$- \mathbb{E}[D_{\mathrm{KL}}(q^{(T)}(\mathbf{x}_T|\mathbf{x}_0, \mathbf{y}) \| p_\theta^{(T)}(\mathbf{x}_T|\mathbf{y}))] \tag{8}$$

where $q(\mathbf{x}_0)$ is the data distribution, $q(\mathbf{y}|\mathbf{x}_0)$ follows Equation 1, the expectation on the right hand side is given by sampling $\mathbf{x}_0 \sim q(\mathbf{x}_0)$, $\mathbf{y} \sim q(\mathbf{y}|\mathbf{x}_0)$, $\mathbf{x}_T \sim q^{(T)}(\mathbf{x}_T|\mathbf{x}_0, \mathbf{y})$, and $\mathbf{x}_t \sim q^{(t)}(\mathbf{x}_t|\mathbf{x}_{t+1}, \mathbf{x}_0, \mathbf{y})$ for $t \in [1, T-1]$.

## C EQUIVALENCE BETWEEN "VARIANCE PRESERVING" AND "VARIANCE EXPLODING" DIFFUSION MODELS

In our main paper, we describe our methods based on the "Variance Exploding" hyperparameters $\sigma_t$, where $\sigma_t \in [0, \infty)$ and

$$q(\mathbf{x}_t|\mathbf{x}_0) = \mathcal{N}(\mathbf{x}_0, \sigma_t^2 \mathbf{I}). \tag{9}$$

In DDIM (Song et al., 2021), the hyperparameters are "Variance Preserving" ones $\alpha_t$, where $\alpha_t \in (0, 1]$ and

$$q(\mathbf{x}_t|\mathbf{x}_0) = \mathcal{N}(\sqrt{\alpha_t}\mathbf{x}_0, (1 - \alpha_t)\mathbf{I}). \tag{10}$$

We use the colored notation $\mathbf{x}_t$ to emphasize that this is different from $\mathbf{x}_t$ (an exception is $\mathbf{x}_0 = \mathbf{x}_0$). Using the reparametrization trick, we have that:

$$\mathbf{x}_t = \mathbf{x}_0 + \sigma_t \epsilon \tag{11}$$

$$\mathbf{x}_t = \sqrt{\alpha_t}\mathbf{x}_0 + \sqrt{1 - \alpha_t}\epsilon \tag{12}$$

where $\epsilon \sim \mathcal{N}(0, \mathbf{I})$. We can divide by $\sqrt{1 + \sigma_t^2}$ in both sides of Equation 11:

$$\frac{\mathbf{x}_t}{\sqrt{1 + \sigma_t^2}} = \frac{\mathbf{x}_0}{\sqrt{1 + \sigma_t^2}} + \frac{\sigma_t}{\sqrt{1 + \sigma_t^2}}\epsilon. \tag{13}$$

Let $\alpha_t = 1/(1 + \sigma_t^2)$, and let $\mathbf{x}_t = \mathbf{x}_t/\sqrt{1 + \sigma_t^2}$; then from Equation 13 we have that

$$\mathbf{x}_t = \sqrt{\alpha_t}\mathbf{x}_0 + \sqrt{1 - \alpha_t}\epsilon, \tag{14}$$

which is equivalent to the "Variance Preserving" case. Therefore, we can use "Variance Preserving" models, such as DDPM, directly in our DDRM updates, even though the latter uses the "Variance Exploding" parametrization:

1. From $\mathbf{x}_t$, obtain predictions $\epsilon$ and $\mathbf{x}_t = \mathbf{x}_t\sqrt{1 + \sigma_t^2}$.
2. From $\mathbf{x}_t$ and $\epsilon$, apply DDRM updates to get $\mathbf{x}_{t-1}$.
3. From $\mathbf{x}_{t-1}$, get $\mathbf{x}_{t-1} = \mathbf{x}_{t-1}/\sqrt{1 + \sigma_{t-1}^2}$.

Note that although the inference algorithms are shown to be equivalent, the choice between "Variance Preserving" and "Variance Exploding" may affect the training of diffusion networks.

## D PROOFS

**Theorem 3.1.** *Assume that the models $f_\theta^{(t)}$ and $f_\theta^{(t')}$ are independent whenever $t \neq t'$, then when $\eta = 1$ and $\eta_b = \frac{2\sigma_t^2}{\sigma_t^2 + \sigma_{\mathbf{y}}^2/s_i^2}$, the ELBO objective of DDRM (details in Equation 8) can be rewritten in the form of the DDPM / DDIM objective (Song et al., 2021).*

*Proof.* As there is no parameter sharing between models at different time steps $t$, let us focus on any particular time step $t$ and rewrite the corresponding objective as a denoising autoencoder objective.

**Case I** For $t > 0$, the only term in Equation 8 that is related to $f_\theta^{(t)}$ (which is used to make the prediction $\mathbf{x}_{\theta,t}$) is:

$$
\begin{aligned}
& D_{\mathrm{KL}}(q^{(t)}(\mathbf{x}_t|\mathbf{x}_{t+1}, \mathbf{x}_0, \mathbf{y}) \| p_\theta^{(t)}(\mathbf{x}_t|\mathbf{x}_{t+1}, \mathbf{y})) \\
= {}& D_{\mathrm{KL}}(q^{(t)}(\bar{\mathbf{x}}_t|\mathbf{x}_{t+1}, \mathbf{x}_0, \mathbf{y}) \| p_\theta^{(t)}(\bar{\mathbf{x}}_t|\mathbf{x}_{t+1}, \mathbf{y})) \\
= {}& \sum_{i=1}^n D_{\mathrm{KL}}(q^{(t)}(\bar{\mathbf{x}}_t^{(i)}|\mathbf{x}_{t+1}, \mathbf{x}_0, \mathbf{y}) \| p_\theta^{(t)}(\bar{\mathbf{x}}_t^{(i)}|\mathbf{x}_{t+1}, \mathbf{x}_0, \mathbf{y})),
\end{aligned}
\tag{15}
$$

where the first equality is from the orthogonality of $\boldsymbol{V}^\top$ and the second equality is from the fact that both $q^{(t)}$ and $p_\theta^{(t)}$ over the spectral space are Gaussians with identical diagonal covariance matrices (so the KL divergence can factorize).

Here, we will use a simple property of the KL divergence between univariate Gaussians (Kingma & Welling, 2013):

If $p = \mathcal{N}(\mu_1, V_1), q = \mathcal{N}(\mu_2, V_2)$, then

$$
D_{\mathrm{KL}}(p\|q) = \frac{1}{2}\log\frac{V_2}{V_1} + \frac{V_1 + (\mu_1 - \mu_2)^2}{2V_2} - \frac{1}{2}.
$$

Since we constructed $p_\theta^{(t)}$ and $q^{(t)}$ to have the same variance, Equation 15 is a total squared error with weights for each dimension of $\bar{\mathbf{x}}_t$ (the spectral space), so the DDPM objective (which is a total squared error objective in the original space) is still a good approximation. In order to transform it into a denoising autoencoder objective (equivalent to DDPM), the weights have to be equal. Next, we will show that our construction of $\eta = 1$ and $\eta_b = 2\sigma_t^2/(\sigma_t^2 + \sigma_\mathbf{y}^2/s_i^2)$ satisfies this.

All the indices $i$ will fall into one of the three cases: $s_i = 0$, $\sigma_t < \sigma_\mathbf{y}/s_i$, or $\sigma_t > \sigma_\mathbf{y}/s_i$.

- For $s_i = 0$, the KL divergence is $\frac{(\bar{\mathbf{x}}_{\theta,t}^{(i)} - \bar{\mathbf{x}}_0^{(i)})^2}{2\sigma_t^2}$, where we recall $\bar{\mathbf{x}}_{\theta,t} = \boldsymbol{V}^\top f_\theta^{(t)}(\mathbf{x}_{t+1})$.

- For $\sigma_t < \frac{\sigma_\mathbf{y}}{s_i}$, the KL divergence is also $\frac{(\bar{\mathbf{x}}_{\theta,t}^{(i)} - \bar{\mathbf{x}}_0^{(i)})^2}{2\sigma_t^2}$.

- For $\sigma_t \geq \frac{\sigma_\mathbf{y}}{s_i}$, we have defined $\eta_b$ as a solution to the following quadratic equation (the other solution is 0, which is irrelevant to our case since it does not make use of information from $\mathbf{y}$):

$$
(\sigma_t^2 + \frac{\sigma_\mathbf{y}^2}{s_i^2})\eta_b^2 - 2\sigma_t^2\eta_b = 0;
\tag{16}
$$

reorganizing terms, we have that:

$$
(\sigma_t^2 + \frac{\sigma_\mathbf{y}^2}{s_i^2})\eta_b^2 - 2\sigma_t^2\eta_b + \sigma_t^2 = \sigma_t^2
$$

$$
\sigma_t^2(1 - \eta_b)^2 = \sigma_t^2\eta_b^2 - 2\sigma_t^2\eta_b + \sigma_t^2 = \sigma_t^2 - \frac{\sigma_\mathbf{y}^2}{s_i^2}\eta_b^2
$$

$$
\frac{(1 - \eta_b)^2}{\sigma_t^2 - \frac{\sigma_\mathbf{y}^2}{s_i^2}\eta_b^2} = \frac{1}{\sigma_t^2},
\tag{17}
$$

So the KL divergence is

$$
\frac{(1 - \eta_b)^2}{2(\sigma_t^2 - \frac{\sigma_\mathbf{y}^2}{s_i^2}\eta_b^2)}(\bar{\mathbf{x}}_{\theta,t}^{(i)} - \bar{\mathbf{x}}_0^{(i)})^2 = \frac{(\bar{\mathbf{x}}_{\theta,t}^{(i)} - \bar{\mathbf{x}}_0^{(i)})^2}{2\sigma_t^2}.
$$

Therefore, regardless of how the cases are distributed among indices, we will always have that:

$$D_{\mathrm{KL}}(q^{(t)}(\bar{\mathbf{x}}_t|\mathbf{x}_{t+1},\mathbf{x}_0,\mathbf{y})\|p_\theta^{(t)}(\bar{\mathbf{x}}_t|\mathbf{x}_{t+1},\mathbf{y})) = \sum_{i=1}^{n^2} \frac{(\bar{\mathbf{x}}_{\theta,t}^{(i)} - \bar{\mathbf{x}}_0^{(i)})^2}{2\sigma_t^2} = \frac{\|\bar{\mathbf{x}}_{\theta,t} - \bar{\mathbf{x}}_0\|_2^2}{2\sigma_t^2} = \frac{\|f_\theta^{(t)}(\mathbf{x}_{t+1}) - \mathbf{x}_0\|_2^2}{2\sigma_t^2}.$$

**Case II** For $t = 0$, we will only have two cases ($s_i = 0$ or $\sigma_t < \frac{\sigma_\mathbf{y}}{s_i}$), and thus, similar to **Case I**,

$$\log p_\theta^{(0)}(\bar{\mathbf{x}}_0|\mathbf{x}_1,\mathbf{y}) = \sum_{i=1}^{n^2} \log p_\theta^{(0)}(\bar{\mathbf{x}}_0^{(i)}|\mathbf{x}_1,\mathbf{y}) \propto \sum_{i=1}^{n^2} (\bar{\mathbf{x}}_{\theta,0}^{(i)} - \bar{\mathbf{x}}_0^{(i)})^2 = \|\bar{\mathbf{x}}_{\theta,0} - \bar{\mathbf{x}}_0\|_2^2 = \|f_\theta^{(0)}(\mathbf{x}_1) - \mathbf{x}_0\|_2^2,$$

as long as we have a constant variance for $p_\theta^{(0)}$. Thus, every individual term in Equation 8 can be written as a denoising autoencoder objective, completing the proof. $\square$

# E    MEMORY EFFICIENT SVD

Here we explain how we obtained the singular value decomposition (SVD) for different degradation models efficiently.

## E.1    DENOISING

In denoising, the corrupted image is the original image with additive white Gaussian noise. Therefore, $\boldsymbol{H} = \boldsymbol{I}$ and all the SVD elements of $\boldsymbol{H}$ are simply the identity matrix $\boldsymbol{I}$, which in turns makes their multiplication by different vectors trivial.

## E.2    INPAINTING

In inpainting, $\boldsymbol{H}$ retains a known subset of size $k$ of the image's pixels. This is equivalent to permuting the pixels such that the retained one are placed at the top, then keeping the first $k$ entries. Therefore,

$$\boldsymbol{H} = \boldsymbol{I}\boldsymbol{\Sigma}\boldsymbol{P}, \tag{18}$$

where $\boldsymbol{P}$ is the appropriate permutation matrix, $\boldsymbol{\Sigma}$ is a rectangular diagonal matrix of size $k \times n$ with ones in its main diagonal, and $\boldsymbol{I}$ is the identity matrix. Since permutation matrices are orthogonal, Equation 18 is the SVD of $\boldsymbol{H}$.

We can multiply a given vector by $\boldsymbol{P}$ and $\boldsymbol{P}^T$ by storing the permutation itself rather than the matrix. $\boldsymbol{\Sigma}$ can multiply a vector by simply slicing it. Therefore, by storing the appropriate permutation and the number $k$, we can apply each element of the SVD with $\Theta(n)$ space complexity.

## E.3    SUPER RESOLUTION

For super resolution, we assume that the original image of size $d \times d$ (*i.e.* $n = 3d^2$) is downscaled using a block averaging filter by $r$ in each dimension, such that $d$ is divisible by $r$. In this scenario, each pixel in the output image is the average of an $r \times r$ patch in the input image, and each such patch affects exactly one output pixel. Therefore, any output pixel is given by $(\boldsymbol{H}\mathbf{x})_i = \boldsymbol{k}^T \boldsymbol{p}_i$, where $\boldsymbol{k}$ is a vector of size $r^2$ with $\frac{1}{r^2}$ in each entry, and $\boldsymbol{p}_i$ is the vectorized $i$-th $r \times r$ patch. More formally, if $\boldsymbol{P}_1$ is a permutation matrix that reorders a vectorized image into patches, then

$$\boldsymbol{H} = \left(\boldsymbol{I} \otimes \boldsymbol{k}^T\right) \boldsymbol{P}_1,$$

where $\otimes$ is the Kronecker product, and $\boldsymbol{I}$ is the identity matrix of size $\frac{d}{r} \times \frac{d}{r}$. In order to obtain the SVD of $\boldsymbol{H}$, we calculate the SVD of $\boldsymbol{k}^T$:

$$\boldsymbol{k}^T = \boldsymbol{U_k}\boldsymbol{\Sigma_k}\boldsymbol{V_k}^T.$$

Using properties of the Kronecker product, we observe

$$\boldsymbol{H} = \left(\boldsymbol{I} \otimes \boldsymbol{k}^T\right) \boldsymbol{P}_1 = \left((\boldsymbol{III}) \otimes \left(\boldsymbol{U_k}\boldsymbol{\Sigma_k}\boldsymbol{V_k}^T\right)\right) \boldsymbol{P}_1 \tag{19}$$

$$= (\boldsymbol{I} \otimes \boldsymbol{U_k}) (\boldsymbol{I} \otimes \boldsymbol{\Sigma_k}) \left(\boldsymbol{I} \otimes \boldsymbol{V_k^T}\right) \boldsymbol{P}_1.$$

The Kronecker product of two orthogonal matrices is an orthogonal matrix. Therefore, $\boldsymbol{I} \otimes \boldsymbol{U_k}$ and $\boldsymbol{I} \otimes \boldsymbol{V_k^T}$ are orthogonal. Observe that the matrix $\boldsymbol{I} \otimes \boldsymbol{\Sigma_k}$ has one non-zero value ($\frac{1}{r^2}$) in each row. By applying a simple permutation on its columns, these values can be reordered to be on the main diagonal. We denote the appropriate permutation matrix by $\boldsymbol{P}_2$, and obtain

$$\boldsymbol{H} = \boldsymbol{U}\boldsymbol{\Sigma}\boldsymbol{V}^T, \tag{20}$$

where $\boldsymbol{U} = \boldsymbol{I} \otimes \boldsymbol{U_k}$ is orthogonal, $\boldsymbol{\Sigma} = (\boldsymbol{I} \otimes \boldsymbol{\Sigma_k}) \boldsymbol{P}_2^T$ is a rectangular diagonal matrix with non-negative entries, and $\boldsymbol{V}^T = \boldsymbol{P}_2 \left(\boldsymbol{I} \otimes \boldsymbol{V_k^T}\right) \boldsymbol{P}_1$ is orthogonal. As such, Equation 20 is the SVD of $\boldsymbol{H}$. By storing the permutations and the SVD elements of $\boldsymbol{k}^T$, we can simulate each element of the SVD of $\boldsymbol{H}$ with $\Theta(n)$ space complexity, without directly calculating the Kronecker products with $\boldsymbol{I}$.

### E.4  COLORIZATION

The grayscale image is obtained by averaging the red, green, and blue channels of each pixel. This means that every output pixel is given by $(\boldsymbol{H}\mathbf{x})_i = \boldsymbol{k}^T \boldsymbol{p}_i$, where $\boldsymbol{k}^T = \left(\frac{1}{3} \quad \frac{1}{3} \quad \frac{1}{3}\right)$ and $\boldsymbol{p}_i$ is the 3-valued $i$-th pixel of the original color image. The SVD of $\boldsymbol{H}$ is obtained exactly the same as in the super resolution case, with separate pixels replacing separate patches.

### E.5  DEBLURRING

We focus on *separable blurring*, where the 2D blurring kernel is $\boldsymbol{K} = \boldsymbol{r}\boldsymbol{c}^T$, which means $\boldsymbol{c}$ is applied on the columns of the image, and $\boldsymbol{r}^T$ is applied on its rows. The blurred image can be obtained by $\boldsymbol{B} = \boldsymbol{A}_c \boldsymbol{X} \boldsymbol{A}_r^T$, where $\boldsymbol{A}_c$ and $\boldsymbol{A}_r$ apply a 1D convolution with kernels $\boldsymbol{c}$ and $\boldsymbol{r}$, respectively. Alternatively, $\boldsymbol{b} = \boldsymbol{H}\boldsymbol{x}$, where $\boldsymbol{x}$ is the vectorized image $\boldsymbol{X}$, $\boldsymbol{b}$ is the vectorized blurred image $\boldsymbol{B}$, and $\boldsymbol{H}$ is the matrix applying the 2D convolution $\boldsymbol{K}$. It can be shown that $\boldsymbol{H} = \boldsymbol{A}_r \otimes \boldsymbol{A}_c$, where $\otimes$ is the Kronecker product. In order to calculate the SVD of $\boldsymbol{H}$, we calculate the SVD of $\boldsymbol{A}_r$ and $\boldsymbol{A}_c$:

$$\boldsymbol{A}_r = \boldsymbol{U}_r \boldsymbol{\Sigma}_r \boldsymbol{V}_r^T, \quad \boldsymbol{A}_c = \boldsymbol{U}_c \boldsymbol{\Sigma}_c \boldsymbol{V}_c^T.$$

Using the properties of the Kronecker product, we observe

$$\begin{aligned} \boldsymbol{H} = \boldsymbol{A}_r \otimes \boldsymbol{A}_c &= \left(\boldsymbol{U}_r \boldsymbol{\Sigma}_r \boldsymbol{V}_r^T\right) \otimes \left(\boldsymbol{U}_c \boldsymbol{\Sigma}_c \boldsymbol{V}_c^T\right) \\ &= (\boldsymbol{U}_r \otimes \boldsymbol{U}_c)(\boldsymbol{\Sigma}_r \otimes \boldsymbol{\Sigma}_c)(\boldsymbol{V}_r \otimes \boldsymbol{V}_c)^T. \end{aligned} \tag{21}$$

The Kronecker product preserves orthogonality. Therefore, Equation 21 is a valid SVD of $\boldsymbol{H}$, with the exception of the singular values not being on the main diagonal, and not being sorted descendingly. We reorder the columns so that the singular values are on the main diagonal and denote the corresponding permutation matrix by $\boldsymbol{P}_1$. We also sort the values descendingly and denote the sorting permutation matrix by $\boldsymbol{P}_2$, and obtain the following SVD:

$$\boldsymbol{H} = \boldsymbol{U}\boldsymbol{\Sigma}\boldsymbol{V}^T, \tag{22}$$

where $\boldsymbol{U} = (\boldsymbol{U}_r \otimes \boldsymbol{U}_c) \boldsymbol{P}_2^T$, $\boldsymbol{\Sigma} = \boldsymbol{P}_2 (\boldsymbol{\Sigma}_r \otimes \boldsymbol{\Sigma}_c) \boldsymbol{P}_1^T \boldsymbol{P}_2^T$, and $\boldsymbol{V}^T = \boldsymbol{P}_2 \boldsymbol{P}_1 (\boldsymbol{V}_r \otimes \boldsymbol{V}_c)^T$.

For every matrix of the form $\boldsymbol{M} = \boldsymbol{N} \otimes \boldsymbol{L}$, it holds that $\boldsymbol{M}x$ is the vectorized version of $\boldsymbol{L}\boldsymbol{X}\boldsymbol{N}^T$. By using this property and applying the relevant permutation, we can simulate multiplying a vector by $\boldsymbol{U}$, $\boldsymbol{V}$, $\boldsymbol{U}^T$, or $\boldsymbol{V}^T$ without storing the full matrix. The space complexity of this approach is $\Theta(n)$, which is required for computing the SVD of $\boldsymbol{A}_r$ and $\boldsymbol{A}_c$, as well as storing the permutations.

## F  ABLATION STUDIES ON HYPERPARAMETERS

$\eta$ **and** $\eta_{\mathbf{b}}$.  Apart from the timestep schedules, DDRM has two hyperparameters $\eta$ and $\eta_b$, which control the level of noise injected at each timestep. To identify an ideal combination, we perform a hyperparameter search over $\eta, \eta_b \in \{0.7, 0.8, 0.9, 1.0\}$ for the task of deblurring with $\sigma_y = 0.05$ in 1000 ImageNet validation images, using the model trained in (Dhariwal & Nichol, 2021). It is possible to also consider different $\eta$ values for $s_i = 0$ and $\sigma_i < \sigma_\mathbf{y}/s_i$; we leave that as future work.

Table 3: Ablation studies on $\eta$ and $\eta_b$.

(a) PSNR ($\uparrow$).

| $\eta$ \ $\eta_b$ | 0.7 | 0.8 | 0.9 | 1.0 |
|---|---|---|---|---|
| 0.7 | 25.16 | 25.19 | 25.20 | 25.20 |
| 0.8 | 25.17 | 25.23 | 25.27 | 25.29 |
| 0.9 | 25.07 | 25.18 | 25.26 | 25.32 |
| 1.0 | 24.54 | 25.75 | 24.91 | 25.04 |

(b) KID $\times 10^3$ ($\downarrow$).

| $\eta$ \ $\eta_b$ | 0.7 | 0.8 | 0.9 | 1.0 |
|---|---|---|---|---|
| 0.7 | 16.27 | 14.30 | 12.76 | 11.65 |
| 0.8 | 21.07 | 19.07 | 17.37 | 15.98 |
| 0.9 | 27.85 | 25.64 | 23.81 | 22.40 |
| 1.0 | 45.10 | 42.50 | 40.10 | 37.84 |

We report PSNR and KID results in Table 3. From the results, we observe that generally (*i*) as $\eta_b$ increases, PSNR increases while KID decreases, which is reasonable given that we wish to leverage the information from **y**; (*ii*) as $\eta$ increases, PSNR increases (except for $\eta = 1.0$) yet KID also increases, which presents a trade-off in reconstruction error and image quality (known as the perception-distortion trade-off (Blau & Michaeli, 2018)). Therefore, we choose $\eta_b = 1$ and $\eta = 0.85$ to balance performance on PSNR and KID when we report results.

**Timestep schedules.** The timestep schedule has a direct impact on NFEs, as the wall-clock time is roughly linear with respect to NFEs (Song et al., 2021). In Tables 5 and 6, we compare the PSNR, FID, and KID of DDRM with 20 or 100 timesteps (with or without conditioning) and default $\eta = 0.85$ and $\eta_b = 1$. We observe that DDRM with 20 or 100 timesteps have similar performance when other hyperparameters are identical, with DDRM (20) having a slight edge in FID and KID.

## G  EXPERIMENTAL SETUP OF DGP, RED, AND SNIPS

Recall that we evaluated DGP (Pan et al., 2020), RED (Romano et al., 2017), and SNIPS (Kawar et al., 2021) on $256 \times 256$ ImageNet 1K images, for the problems of $4\times$ super resolution and deblurring. OneNet (Rick Chang et al., 2017) is not included in the comparisons as it is limited to images of size $64 \times 64$, and generalization to higher dimensions requires an improved network architecture. Below we expand on the tested methods' experimental setup.

In each of the inverse problems we show, pixel values are in the range $[0, 1]$, and the degraded measurements are obtained as follows: (*i*) for super-resolution, we use a block averaging filter to downscale the images by the same factor in each axis; (*ii*) for deblurring, the images are blurred by a $9 \times 9$ uniform kernel; (*iii*) for colorization, the grayscale image is an average of the red, green, and blue channels of the original image; (*iv*) and for inpainting, we mask parts of the original image with text overlay or randomly drop $50\%$ of the pixels. Additive white Gaussian noise can optionally be added to the measurements in all inverse problems.

For DGP (Pan et al., 2020), we use the same hyperparameters introduced in the original paper for MSE-biased super resolution. We note that the downscaling applied in DGP is different from the block averaging filter that we used, and the numbers they reported are on the $128 \times 128$ resolution. Nevertheless, in our experiments, DGP achieved a PSNR of 23.06 on ImageNet 1K $256 \times 256$ block averaging $4\times$ super resolution, which is similar to the 23.30 reported in the original work. When applied on the deblurring problem, we retained the same DGP hyperparameters as well.

For RED (Romano et al., 2017), we apply the iterative algorithm only in the luminance channel of the image in the YCbCr space, as done in the original paper for deblurring and super resolution. As for the denoising engine enabling the algorithm, we use the same diffusion model used in DDRM to enable as fair a comparison as possible. We use the last step of the diffusion model (equivalent to denoising with $\sigma = 0.005$), as we found it to work best empirically. We also chose the steepest-descent version (RED-SD), and $\lambda = 500$ for best PSNR performance given the denoiser we used. We also set $\sigma_0 = 0.01$ when the measurements are noiseless, because $\sigma_0$ cannot be 0 as RED divides by it.

In super resoltion, RED is initialized with the bicubic upsampled low-res image. In deblurring, it is initialized with the blurry image. We then run RED on the ImageNet 1K for different numbers of steps (see Table 4), and choose the best PSNR for each problem. Namely, we show in our paper RED

Table 4: RED results on ImageNet 1K ($256 \times 256$) for $4\times$ super resolution and deblurring for different numbers of steps.

| | Super-res | | Deblurring | |
|---|---|---|---|---|
| Steps | PSNR↑ | KID↓ | PSNR↑ | KID↓ |
| 0 | 25.65 | 44.90 | 19.26 | 38.00 |
| 20 | 26.05 | 52.51 | 23.49 | 21.99 |
| 100 | 26.08 | 53.55 | 25.00 | 26.09 |
| 500 | 26.00 | 54.19 | 26.16 | 21.21 |

Table 5: ImageNet 50K validation set ($256 \times 256$) results on $4\times$ super resolution with additive noise of $\sigma_\mathbf{y} = 0.05$.

| Method | PSNR↑ | FID↓ | KID↓ | NFEs↓ |
|---|---|---|---|---|
| Bicubic | 22.65 | 64.24 | 50.56 | **0** |
| DDRM | 24.70 | 20.16 | 15.25 | 100 |
| DDRM-CC | **24.71** | 18.22 | 13.57 | 100 |
| DDRM | 24.29 | 17.88 | 13.18 | 20 |
| DDRM-CC | 24.30 | **15.92** | **11.47** | 20 |

on super resolution with 100 steps, and on deblurring with 500 steps. Interestingly, RED achieves a PSNR close to its best for super resolution in just 20 steps. However, DDRM (with 20 steps) still outperforms RED in PSNR, with substantially better perceptual quality (see Table 1).

SNIPS (Kawar et al., 2021) did not originally work with ImageNet images. However, considering the method's similarity to DDRM (as both operate in the spectral space of $\boldsymbol{H}$), a comparison is necessary. We apply SNIPS with the same underlying diffusion model (with all 1000 timesteps) as DDRM for fairness. SNIPS evaluates the diffusion model $\tau$ times for each timestep. We set $\tau = 1$ so that SNIPS' runtime remains reasonable in comparison to the rest of the considered methods, and do not explore higher values of $\tau$. It is worth mentioning that in the original work, $\tau$ was set to 3 for an LSUN bedrooms diffusion model with 1086 timesteps. We set $c = 0.67$ as it achieved the best PSNR performance.

The original work in SNIPS calculates the SVD of $\boldsymbol{H}$ directly, which hinders its ability to handle $256 \times 256$ images on typical hardware. In order to draw comparisons, we replaced the direct calculation of the SVD with our efficient implementation detailed in Appendix E.

In Figure 3 and Table 2, we show that DGP, RED, and SNIPS all fail to produce viable results when significant noise is added to the measurements. For these results, we use the same hyperparameters used in the noiseless case for all algorithms (except $\sigma_\mathbf{y}$ where applicable). While tuning the hyperparameters may boost performance, we do not explore that option as we are only interested in algorithms where given $\boldsymbol{H}$ and $\sigma_\mathbf{y}$, the restoration process is automatic. To further demonstrate DDRM's capabilities and speed, we evaluate it on the entire $50,000$-image ImageNet validation set in Tables 5 and 6, reporting Fréchet Inception distance (FID; Heusel et al. (2017)) as well as KID, as enough samples are available.

We used the same hyperparameters for noisy and noiseless versions of the same problem for DGP, RED, and SNIPS, as tuning them for each version would compromise their unsupervised nature.

Table 6: ImageNet 50K validation set ($256 \times 256$) results on deblurring with additive noise of $\sigma_{\mathbf{y}} = 0.05$.

| Method | PSNR↑ | FID↓ | KID↓ | NFEs↓ |
|---|---|---|---|---|
| Blurry | 18.05 | 93.36 | 74.13 | 0 |
| DDRM | 24.23 | 22.30 | 16.23 | 100 |
| DDRM-CC | 24.21 | 20.06 | 14.20 | 100 |
| DDRM | 24.60 | 21.60 | 15.65 | 20 |
| DDRM-CC | **24.61** | **19.66** | **13.94** | 20 |

