# OpenReview forum: "Denoising Diffusion Restoration Models"
_ICLR.cc/2022/Workshop/DGM4HSD — ICLR 2022 DGM4HSD workshop Oral_

### Official Review · Reviewer_Pwi9 · 2022-03-19
**Well-formulated problem with good results.**

**Rating:** 8
**Confidence:** 3

**Review:**

The paper introduces Denoising Diffusion Restoration Model (DDRM) which is an unsupervised posterior sampling method to solve linear inverse problems in image restoration. In my opinion the paper is well written and formulated, producing interesting results and generalisability capacity across a varied range of tasks.

My comments are as follows:
- In Introduction the authors mention that DDRM outperforms previous methods in several metrics where runtime is included. Where are the experimental results to back this claim on runtime?
- A main claim of the paper is that DDRM solves the previous dilemma of inverse problem solvers in which unsupervised approaches are inefficient and supervised approaches can only address specific problems. In this sense, DDRM is claimed to be an unsupervised solver. However, from the formulation explained in section 3, doesn't the existence of both signal x and measurements y in the model specification, mean that DDRM is a supervised model? What am I missing here?
- Do the authors have any hypothesis on the strange result in Table 1, for KID (Deblurring) when using SNIPS? The value seems weirdly off.
- In section 4, I'm concerned by the sentence "DGP and DDRM-CC use ground-truth class labels for the test images to aid in the restoration process, and thus have an unfair advantage.". If it is the case that these methods use ground-truth labels, doesn't this mean these models cannot be used in practice and therefore should just not be used? From tables 1 and 2, DDRM-CC indeed achieves good results, but based on this sentence I feel its results are therefore a bit meaningless in practice.

---

### Official Review · Reviewer_tYrr · 2022-03-25

**Rating:** 7
**Confidence:** 4

**Review:**

The authors format classical image restoration linear inverse problems (IIP) and propose the denoising diffusion restoration models (DDRM) as an unsupervised solution for the IIP.

***

Overall the question and method are clear, one downside is that the paper format is missing the abstract section - I think maybe it is due to the page constraints but it is still a bit weird. The authors may consider removing Table 2 to the appendix with a hyperlink.

Some additional questions could be if pre-trained models coming from a different domain would affect the DDRM or not; sample and time complexity of DDRM compared to some recent methods. (Song  et  al.,  2021)

---

### Official Review · Reviewer_wwoU · 2022-03-28
**Good paper that proposes a variational inference approach to sampling**

**Rating:** 8
**Confidence:** 4

**Review:**

This paper considers the problem of sampling from the posterior distribution of a score-based model with applications to inverse problems. In this case, the measurements are $y = Ax^* + $ noise, and one wishes to sample from $p(x|y)$, given access to a score-based model that can compute $\nabla \log p_\sigma(x) $, where $p_\sigma$ is the molification of a density $p$ by a Gaussian of variance $\sigma^2$.

Sampling from such a distribution is difficult since the molification noise can interfere with the noise in the measurements. Prior work resolved this using an SVD of the matrix $A$, but was very slow. This work proposes a faster sampling algorithm using variational inference.

Positives:
+ The empirical results are very good, as are the numerical results.
+ The algorithm and analysis have reasonably high technical quality.
+ The work is novel and significant, as a lot of recent research has considered faster sampling algorithms for score-based models.
+ The writing is generally good.

Negatives:
- I am a little confused about the reason for the performance improvement. While the paper starts from the perspective of variational inference, eventually the algorithm becomes SNIPS (which has been cited in the submission). So does the performance benefit come purely from the fact that some diffusion happens in the spectral space of the operator, and this paper establishes a connection between SNIPS and variational inference? Or is there something more to it? The paper could use a discussion of this, or an appropriate ablation study.
- I think some compressed sensing experiments can improve the experimental results a little bit. In compressed sensing, we know there exist cases where we can (almost) exactly recover the ground truth signal, and comparison with state-of-the-art in compressed sensing would give more insight over lossy cases like inpainting.

---

### Decision · Program_Chairs · 2022-03-27

Accept (Oral)